# Diversity of Bacterial Biosynthetic Genes in Maritime Antarctica

**DOI:** 10.3390/microorganisms8020279

**Published:** 2020-02-18

**Authors:** Adriana Rego, António G. G. Sousa, João P. Santos, Francisco Pascoal, João Canário, Pedro N. Leão, Catarina Magalhães

**Affiliations:** 1Interdisciplinary Centre of Marine and Environmental Research (CIIMAR), University of Porto, 4450-208 Matosinhos, Portugal; adriana.rego@ciimar.up.pt (A.R.); antonio.sousa@ciimar.up.pt (A.G.G.S.); joaofs21@gmail.com (J.P.S.); fpascoal1996@gmail.com (F.P.); 2Institute of Biomedical Sciences Abel Salazar (ICBAS), University of Porto, 4050-313 Porto, Portugal; 3Institute F.-A. Forel, Earth and Environmental Sciences, Faculty of Sciences, University of Geneva, 66, Boulevard Carl-Vogt, 1211 Genève 4, Switzerland; 4Centro de Química Estrutural at Instituto Superior Técnico, Universidade de Lisboa, Av. Rovisco Pais, 1049-001 Lisboa, Portugal; joao.canario@tecnico.ulisboa.pt; 5Faculty of Sciences, University of Porto, 4150-179 Porto, Portugal; 6School of Science, University of Waikato, Hamilton 3216, New Zealand

**Keywords:** Antarctica, polyketides (PKs), non-ribosomal peptides (NRPs), biosynthetic genes, computational bioprospection, ketosynthase (KS), adenylation (AD), natural products (NPs)

## Abstract

Bacterial natural products (NPs) are still a major source of new drug leads. Polyketides (PKs) and non-ribosomal peptides (NRP) are two pharmaceutically important families of NPs and recent studies have revealed Antarctica to harbor endemic polyketide synthase (PKS) and non-ribosomal peptide synthetase (NRPS) genes, likely to be involved in the production of novel metabolites. Despite this, the diversity of secondary metabolites genes in Antarctica is still poorly explored. In this study, a computational bioprospection approach was employed to study the diversity and identity of PKS and NRPS genes to one of the most biodiverse areas in maritime Antarctica—Maxwell Bay. Amplicon sequencing of soil samples targeting ketosynthase (KS) and adenylation (AD) domains of PKS and NRPS genes, respectively, revealed abundant and unexplored chemical diversity in this peninsula. About 20% of AD domain sequences were only distantly related to characterized biosynthetic genes. Several PKS and NRPS genes were found to be closely associated to recently described metabolites including those from uncultured and candidate phyla. The combination of new approaches in computational biology and new culture-dependent and -independent strategies is thus critical for the recovery of the potential novel chemistry encoded in Antarctica microorganisms.

## 1. Introduction

Microorganisms, and bacteria in particular, produce a plethora of bioactive natural products (NPs), including clinically-relevant antibiotics, antifungals and anticancer agents, and continue to be a major source of new structural leads [1]. Despite the vast structural diversity of these bacterial small molecules, the majority of biomedically relevant NPs belong to the polyketide (PK) or non-ribosomal peptide (NRP) biogenetic families, or to their hybrids [2]. PKs and NRPs are biosynthesized by polyketide synthase (PKS) and non-ribosomal peptide synthetase (NRPS) enzymatic machinery, respectively [3]. Conserved motifs on the genes encoding these enzymes have allowed for the design of degenerate primers [4,5], that have proved useful to determine PKS and NRPS phylogenetic diversity in natural ecosystems [6], and also to predict (to some extent) their encoded chemical outputs [7]. Such primers target the highly conserved ketosynthase (KS) and adenylation (AD) domains of PKS and NRPS genes, respectively. A growing number of studies have employed the PCR-based sequence tag approach to uncover PKS and NRPS gene diversity across different environmental microbiomes [7,8,9].

Extreme environments, such as Antarctica, have the potential to reveal novel molecules since they are still a poorly explored source of chemical scaffolds and novel chemical diversity [10,11]. As such, metabarcoding approaches to study PKS and NRPS diversity in Antarctic ecosystems represent a simple and effective strategy to understand the biosynthetic potential of a large number of sampling sites [7]. A couple of such studies have been carried out very recently in Antarctica [12,13] revealing that soils at different Antarctica locations harbor multiple endemic PKS and NRPS genes, likely to be involved in the synthesis of new chemical diversity. In Antarctic soils, the percentage of yet-uncultured genera is expected to be much higher [14] than the 82% described for general soils [15]. A considerable fraction of such uncultured bacteria corresponds to members of the NP-rich Actinobacteria [16,17] and Cyanobacteria [18,19] phyla.

King George Island is one of the largest ice-free and biodiverse areas in maritime Antarctica [20]. It harbors a high density of scientific stations, particularly in Maxwell Bay [21], where most of terrestrial biology research takes place. Although bacterial diversity studies have been conducted in this area [22,23], the diversity of bacterial biosynthetic genes has not been a target of study. Here, we perform a detailed study on soil samples from Maxwell Bay, with the main goal of assessing the diversity and identity of key biosynthetic genes (PKS and NRPS) and to what extent these are likely to be unique to this peninsular environment.

## 2. Materials and Methods

### 2.1. Sampling Area and Environmental Variables

The study site—Fildes Peninsula—is located on the southwestern part of King George Island, maritime Antarctica (Figure 1). Despite being the subject of intense human activities, the Fildes Peninsula is recognized as a special ecological protection area [21]. Samples were collected during the CONTANTARC-3 campaign in Maxwell Bay, between 5 and 15 February 2014. Surface soil samples (up to 10 cm deep) from eight stations (Figure 1, Appendix A) were collected using a sterile plastic spatula and stored immediately in sterile plastic bags with zip closure [24] and preserved at −20 °C until arriving at our laboratory (CIIMAR, University of Porto, Portugal) where samples were preserved at −80 °C until further analysis. Organic matter (OM) and water content were measured according with previous described methods [25]. Principal component analysis (PCA) was applied to the environmental variables measured in this study and also presented in a previous study [24], using R v. 3.6.1. 

### 2.2. DNA Extraction and 16S rRNA Gene Amplification and Sequencing

DNA was extracted from 0.5 g (w.w.) of each homogenized soil sample, using the PowerSoil™ DNA isolation kit (MoBio, Carlsbad, CA, USA), according to the manufacturer’s instructions. The 16S rRNA gene was amplified with the prokaryotic primer pair 515F-Y (5′-GTGYCAGCMGCCGCGGTAA-3′) [31] and 926R (5′-CCGYCAATTYMTTTRAGTTT-3′) [32] in order to be paired-end sequenced on an Illumina MiSeq platform using V3 Chemistry (Illumina), according to Sousa and co-workers [33]. Sequencing was carried out by LGC Genomics (LGC Genomics GmbH, Berlin, Germany).

### 2.3. Amplification and Sequencing of KS and AD Domain Sequences

Degenerate primer pairs degKS2F (5′-GCNATGGAYCCNCARCARMGNVT)/degKS2R(5′-GTNCCNGTNCCRTGNSCYTCNAC) [5] and A3F (5′-GCSTACSYSATSTACACSTCSGG)/A7R(5′-SASGTCVCCSGTSCGGTA) [4] were used to amplify KS and AD domains, respectively, from the same DNA samples used for the 16S rRNA gene amplification. The PCR reaction was prepared in a volume of 20 μL containing 1× TaKaRA PCR Buffer (TAKARA BIO INC, Shiga, Japan), 1.5 mM MgCl_2_ (TAKARA BIO INC, Shiga, Japan), 250 μM dNTPs (TAKARA BIO INC, Shiga, Japan)), 0.625 μL of each primer (100μM), 0.25 mg/mL of UltraPureTM BSA (Life technologies, Waltham, MA USA), 0.5 U TaKaRa Taq™ Hot Start Version (TAKARA BIO INC Shiga, Japan)), and 2 μL of template DNA. The PCR conditions were executed as following: initial denaturation step at 95 °C for 4 min, followed by 40 cycles of a denaturation step at 94 °C during 40 s, annealing at 56.3 °C for 40 s (KS), or 67.5 °C for 30 s (AD), and extension at 72 °C for 75 s and 60 s, for KS and AD, respectively, followed by a final extension step at 72 °C for 5 min. Amplified PCR products were sequenced using Illumina MiSeq 2×300 technology, as described previously [34], at LGC Genomics (LGC Genomics GmbH, Berlin, Germany).

### 2.4. Sequence Analysis and Taxonomic Assignment of 16S rRNA Gene Amplicons

The sequence analysis of 16S rRNA gene amplicons was performed in DADA2 version 1.12.1 [35] The demultiplexed and primer-clipped reads were trimmed based on quality and length (240 and 160 bp for forward and reverse reads, respectively). Trimmed sequences were dereplicated, denoised, merged, and chimeras were removed. The resulting 16S rRNA gene amplicon sequence variants (ASVs) were taxonomically classified against SILVA v.132 using the Naive Bayes classifier [36]. The ASV table and taxonomy were imported into the phyloseq R package [37] to perform alpha and beta-diversity analysis.

### 2.5. Sequence Analysis and Taxonomic Assignment of KS and AD Domains

Primer-clipped forward and reverse fastq sequences from KS and AD domains were quality trimmed using bbduk, which is part of the BBMap suite version 38.34 (available online: https://sourceforge.net/projects/bbmap/) and truncated to 240 and 175 bp, respectively, using USEARCH v11.0.667 [38], as described in literature [34]. The reads were then reordered to obtain the correct match pairs (using the repair.sh tool from BBMap, available online: https://sourceforge.net/projects/bbmap/) and the matching pairs were concatenated with an intervening “N” using USEARCH. The sequence identifiers of each sample were renamed to allow for discrimination on the statistical analysis and all the samples were combined in a single file. The sequences were dereplicated using USEARCH, clustered at 97% identity, the singletons were removed, and a second round of clustering at 95% of identity was performed, as described previously [34]. Finally, VSEARCH v2.10.4 [39] was used to generate an OTU (operational domain unit) table that was imported to phyloseq [37] for alpha and beta-diversity analysis. The alpha-diversity metrics computed were the number of observed OTUs and Shannon, for beta-diversity, the Bray–Curtis metric was estimated and visualized through principal coordinate analysis method (PCoA). The final plots were obtained using the ggplot2 v.3.1.0 [27] R package. For correlations between the 16S rRNA gene, the most abundant bacterial phyla and the biosynthetic domains, Chao dissimilarity matrices and Mantel tests were computed using the vegan R package 2.56 [40].

For the annotation of KS and AD sequences, the representative OTU sequences generated through USEARCH were aligned locally by BLAST+ version 2.9.0 [41] against the RefSeq non-redundant protein sequences (nr) database. The best hit of each representative KS and AD OTU was retrieved, and BLAST results were curated in order to detect false positives, in which case these were removed. The accession number of each best hit was used to recover the lineage of the closest representatives of each KS and AD OTU using an in-house python script (available online: https://github.com/antonioggsousa/get_taxonomy_lineage.py). The taxonomy classification was included in the respective KS and AD OTUs tables generated in VSEARCH and imported to phyloseq to compute relative abundance of the taxonomic provenience of KS and AD domains that were represented in bar plots.

### 2.6. Phylogenetic Analysis

Multiple sequence alignment (using the ClustalW algorithm) and phylogenetic analysis were performed in MEGA X [42]. The corrected Akaike information criterion (AICc) was used to determine the best nucleotide substitution model in MEGA X. The phylogenetic trees [37] were reconstructed using the maximum likelihood statistical method, bootstrap (with 500 replications), and the corresponding best nucleotide substitution model (GTR + G + I for both domains). Phylogenetic trees were visualized and annotated in iTOL v.4.4.2 [43].

### 2.7. Functional Annotation of KS and AD Domain Sequences to the MiBIG Database

KS and AD OTU sequences were aligned locally with BLAST+ version 2.9.0 against the MiBiG database v.1.4 [44]. BLAST matches with an e-value above 10^−20^ were reported as Not Assigned (NA), due to low statistical confidence. The R packages phyloseq and ggplot2 [27] were used for downstream analysis and visualization, including relative and total abundance classification and functional summary charts. Circos software [45] was used to create heatmaps and circular plots. Chemical structures were drawn using ChemDraw v19.0.1.28 (available online: https://www.perkinelmer.com/category/chemdraw).

## 3. Results and Discussion

### 3.1. Diversity of 16S rRNA Gene, KS and AD Domains in the Maxwell Bay Peninsula, Antarctica

In total, 1710 and 1012 OTUs were successfully retrieved for KS and AD domain of PKS and NRPS genes, respectively. The degenerated primers used showed specificity to the target domains since false positives accounted for less than 4 % and 1% of the total number of OTUs retrieved for KS and AD domains, respectively. Rarefaction plots revealed the diversity of the samples was exhaustively recovered for all samples, except for sample 54 (Appendix A).

To examine differences between 16S rRNA gene diversity and biosynthetic domain diversity between samples, diversity indices (number of ASVs/OTUs and Shannon index) were computed (Figure 2 and Figure 3). A distinct clustering and diversity pattern between the 16S rRNA gene and the conserved biosynthetic motifs of PKS and NRPS genes was detected through the beta-diversity and alpha-diversity metrics (Figure 2). The most diverse samples regarding the distribution of the biosynthetic domains—samples 72, I1, and TC3—were the least diverse for the 16S rRNA gene (Figure 2). It is interesting to note that while computing KS and AD domain alpha-diversity individually for the main prolific phyla (Actinobacteria, Cyanobacteria, and Proteobacteria [46]), it is clear that these phyla are highly diverse in samples 72, I1, and TC3 (Figure 3), which share environmental attributes (Appendix A). To determine the existence of a correlation between the diversity of the functional genes and specific bacterial phyla, Mantel correlation tests between the generated Chao dissimilarity matrices for each marker gene and the bacterial phyla, were performed. A positive correlation between the diversity of biosynthetic genes and the diversity of the known-producers Proteobacteria, Actinobacteria, and Planctomycetes and for the less known Acidobacteria and Gemmatimonadetes bacterial phyla was observed (Table 1). Recently Borsetto and co-workers [13] also found the Gemmatimonadetes phylum to be positively correlated with biosynthetic genes’ diversity.

Curiously, samples 72 and TC3, the most biosynthetically diverse (Figure 2), are located geographically very closely to sample 54 (Figure 1), the least biosynthetically diverse (Figure 2). The beta-diversity metric supports this observation, since sample 54 groups independently, suggesting a distinct biosynthetic composition. In agreement, the principal components analysis (PCA) applied to the environmental variables (Appendix A) indicates that sample 54 was distinguished from others by being associated with high total organic matter and water content (Appendix A). A previous study from our team [47] has revealed that in samples with higher water availability, from McMurdo Dry Vallleys, Antarctica, Actinobacteria dominance was replaced by other phyla, such as Cyanobacteria. Similarly, sample 54 harbors a lower relative abundance and richness of Actinobacteria (Figure 4A) which might be related to the observed shift in KS/AD diversity (Figure 2). Recently Benaud and co-workers [12] reported that drier polar soils are usually associated with a greater amplification of NRPS and PKS genes, which is in agreement with our data.

Previous studies have revealed that soil type [48], actinobacteria richness, geographic location [7], and, more recently, latitude and pH [9] are preponderant factors determining the biosynthetic diversity in environmental microbiomes. In our samples, the geographic location seems to differentiate sample AC3 from the others with respect to environmental characteristics (Appendix A) but did not dictate differences in biosynthetic or taxonomic diversity (Appendix A).

The bacterial taxonomy profile is quite similar across the studied samples, what is expected since samples were collected in a limited spatial area. However, a few changes are noticed such as a relative increased of Proteobacteria and Bacteroidetes in sample TR1 in detriment of Actinobacteria and Verrucomicrobia (Figure 4A). According to the PCA analysis (Appendix A), TR1 diverges from the other samples by being related with higher concentrations of Hg, Pb, Zn, and Cd in soils, thus these chemical variables may be driving differences in the taxonomic composition of this sample. The bacterial community distribution, including the most abundant bacterial phyla (Proteobacteria, Bacteroidetes, Acidobacteria, and Actinobacteria) are in agreement with previous reports in Fildes Peninsula [20,23].

In comparison with similar studies in different ecosystems, such as American desert soil [8], urban park soils [34], and even from soils distributed at a global scale [7] the biosynthetic diversity observed in Maxwell Bay (Appendix A) is clearly inferior, which is expected for a continent with extreme environmental constraints [13]. Very recently, Benaud [12] and Borsetto [13] have assessed biosynthetic diversity in Antarctica, and while diversity indices presented here are not directly comparable with Benaud and co-worker’s study [12] due to different methodological procedures, they are in agreement with the ones presented by Borsetto [13].

The conserved regions of the KS domain are informative of genes tightly associated with bioactivity [49] and allow for a phylogeny-based classification of the PKS gene [50], while this is not observed with the AD domain of NRPSs. A phylogenetic tree constructed from the KS domain sequences obtained in this study (Figure 5), indicates that these sequences cluster within the described classes in the NaPDoS database [50]. The sequences are mainly distributed within modular and hybrid PKS whereas only two OTUs were identified as PUFA synthases and none were associated to the biologically active enediyne class. In the phylogenetic trees for both domains (Figure 5 for KS and Appendix A for AD domains), the sequences from the same sample do not group into a single biosynthetic class or phylogenetic clade, but are distributed along the tree, as previously observed in other studies [8]. These results suggest that the retrieved KS and AD sequences from the different samples share diverse ancestors. It is well-established that PKS and NRPS evolution is driven by horizontal gene transfer (HGT) [46,51], and this has recently been observed also for polar soils [12].

### 3.2. Taxonomic Provenience of the Recovered Biosynthetic Genes’ Sequences

The comparison between 16S rRNA and biosynthetic genes’ diversity taxonomy can provide clues on the ecological roles of the bacterial phyla associated to the biosynthetic diversity and understand their distribution in the bacterial community.

In this study, no congruence was observed between the most representative phyla according to the 16S rRNA gene and the inferred taxonomic provenience for the KS or AD domains (Figure 4). A similar pattern was previously observed by Reddy and co-workers [8] where the metagenomes assessed shared a similar distribution of the major bacterial phyla, however they contained highly distinct collections of secondary metabolite biosynthetic genes. For example, in our data Chlorobi is represented as one of the 10 most abundant bacterial phyla according to the 16S rRNA gene (Figure 4A), however it was not recovered as a major group either for KS (Figure 4B) or AD domains (Figure 4C). The opposite was observed for the Firmicutes phylum, scarcely represented in the 16S rRNA gene dataset (Figure 4A), and predominating in both KS and AD domains (Figure 4B,C). This phylum is known to contain species rich in PKS and/or NRPS biosynthetic gene clusters [46]. A detailed description of the taxonomic provenience of PKS and NRPS genes recovered in the different samples is described below.

#### 3.2.1. Taxonomic Identity of PKS Genes

Proteobacteria is less represented in the KS domain than in 16S rRNA gene (Figure 4A,B), while the relative abundance of Actinobacteria, Cyanobacteria, and Firmicutes increased in the KS domain dataset (Figure 4A,B). Proteobacteria is also considered a NP-rich phylum but the fact that we used primers originally designed to amplify PKS and NRPS genes from Actinobacteria species [4,5], might explain this difference. Interestingly, Gemmatimonadetes was detected in all samples and its abundance was highly represented in the KS domain comparing with the 16S rRNA dataset (Figure 4A,B). This is particularly evident for AC3 and I1 samples, in which KS domains assigned to Gemmatimonadetes account for 14% and 19% of all sequences, respectively (Figure 4B). The same was observed for Verrumicrobia, with 14% and 19% KS domains in samples 66 and TC3, respectively (Figure 4B). Very recently, Gemmatimonadetes and Verrumocrobia have been highlighted in a study revealing novel soil bacteria as harboring diverse genes for natural products’ biosynthesis [52]. OTUs closely related to KS domain sequences from “*Candidatus* Melainabacteria” (closely related to the biosynthetically rich Cyanobacteria) and “*Candidatus* Rukobacteria” were detected in samples I1 and 66 (data not shown, respectively, but no NPs have been reported from these sources).

KS domain provenience was also inspected at lower taxonomic levels to provide information of order and genus associated with the biosynthesis of PKs. Regarding Cyanobacteria, the order Nostocales is widely distributed among the samples and the order Oscillatoriales accounted for 72% and 58% of relative abundance of KS provenience in samples AC3 and I1, respectively (Figure 6C). Oscillatoriales produce the largest number of cyanobacterial metabolites described to date, followed by Nostocales [53]. Interestingly, KS domains sequences identified with high identity to the Cyanobacteria genus *Moorea*, the most prolific genus of cyanobacterial NPs [53], was detected with high abundance in samples AC3 and I1 (Appendix A). Although, to our knowledge there are no reports of Antarctic *Moorea* strains to date, this result might indicate that the recovered sequences are associated with *Moorea* or other closely related genera. KS domains closely associated with the genus *Nostoc,* commonly retrieved from culture-based studies in Antarctica [54], and the second most NP-rich cyanobacterial genus [53], were detected in all samples (Appendix A).

Regarding Actinobacteria, the chemically rich order Strepromycetales is widely distributed and accounts for 58% of KS domains sequences in sample 54 (Figure 6C). Sample TR1 is composed of 88% Corynebacteriales order, whereas sample I1 is composed of 64% of Pseudonocardiales (Figure 6C). Sequences matching to KS domains from the rare genera *Actinoalloteichus, Kitasatospora, Mycobacterium*, and *Rhodococcus* and to the two most NP-rich genera, *Streptomyces* and *Micromonospora* [55], were detected in high abundance across all samples (Appendix A).

Matches to enzymes involved in the production of PKs from Firmicutes belong mostly to the order Bacillales, known to harbor genera rich in secondary metabolite gene clusters [56], however the orders Clostridiales, Lactobacillales, and Selenomonadales were also present (Figure 6E). The genus *Paenibacillus*, commonly retrieved in Antarctica studies [57], is widely distributed across the samples and in sample TR1 is associated with 94% of the KS domains (Appendix A).

Regarding Verrucomicrobia phylum, genus *Chthoniobacter* and *Verrucomicrobium* are well represented across the samples (Appendix A). In fact, recent genomic studies revealed that this phylum harbors biosynthetic gene clusters (BGCs) of PKS origin [58], however their products remain uncharacterized, mostly due to the difficulty of cultivating these bacteria in the laboratory.

A large percentage (from 19% to 59% of relative abundance by sample) of the assembled KS OTUs were most closely related to environmental sample sequences, likely obtained from metagenome surveys as well as biosynthetic gene mining studies across the globe [13,59,60].

#### 3.2.2. Taxonomic Identity of NRPS Genes

The NP-rich Actinobacteria and Cyanobacteria [46] were most closely associated with 50% and 39% of ADs in samples TR1 and 54, respectively (Figure 4C). Proteobacteria-associated AD domain abundance was high across all samples, in this case in agreement with the 16S rRNA gene abundance (Figure 4A,C). The relative abundance of uncultured bacteria is not as high as that observed for KS domains. Additionally, in contrast with the KS domain dataset, several AD domain sequences were associated with Fungi and Planctomycetes, known to be rich in NRPS gene clusters [61,62] (Figure 4C).

At lower taxonomic levels, AD domain sequences associated with the actinobacterial Streptomycetales order were found to be widely distributed across most samples (exception was sample TR1), with the order Corynebacteriales accounting for over 90% of the AD domain closest matches (Figure 6B). Members of the Streptomycetales, specifically the *Streptomyces* genus are responsible for the synthesis of most of the antibiotics currently in use [63]. However, there are few reports of compounds isolated from Antarctic *Streptomyces* strains [64,65,66]. AD domain sequences associated with the non-*Streptomyces* actinobacteria, showed a prevalent distribution (>50% of relative abundance) in some samples: e.g., *Mycobacterium* in sample 87 and *Nocardia* in sample 72 (Appendix A). *Nocardia* strains are also very rich producers of bioactive NPs [67] and have been previously isolated in Antarctic studies [68], but, to our knowledge, no compounds have been reported from Antarctic *Nocardia* strains (Appendix A).

Among Cyanobacteria, the orders Chroococales, Nostocales, and Oscillatoriales were associated with AD domains in all samples (Figure 6D). The most abundant association was to the order Chroococales (45% of relative abundance) and, at the genus level, to *Plankothrix* (Oscillatoriales) (Appendix A).

AD domains associated with the Firmicutes relate almost exclusively to the order Bacillales except in samples 66 and TR1, which harbor a higher relative abundance of members of the Clostridiales (Figure 6F). *Bacillus*, *Brevibacillus*, *Paenibacillus*, and *Tumebacillus* genera are widely distributed across all samples (except *Brevibacillus*, which is absent in sample TR1) (Appendix A).

### 3.3. Metabolites Assignment to KS and AD Domains

One of the ultimate goals of geographically mapping NP biosynthesis genes is to guide the isolation of bioactive compounds [69]. The identification of promising sampling sites, rich in biosynthetic genes encoding for pharmacologically relevant metabolites will ultimately aid the discovery of novel NPs. In this study, 210 and 182 different BGCs from the MiBIG database were assigned to KS and AD domain OTUs, respectively. Sequences that matched with low confidence (e-value above 10^−20^) were not assigned. The percentage of such unassigned OTUs corresponded to 11% and 20% of OTUs from KS and AD domains, respectively. Still, even for the assigned KS and AD domains, this classification is intended only to provide a tentative subclass of PKs or NRPs or likely substructure that might be associated with a particular domain, rather than proof of the presence of a particular compound in the studied sample.

When comparing both genes, it is clear that for the AD domain, a larger percentage of the OTUs were not assigned, particularly for samples 54 and TC3 (Figure 7). Recently, Borsetto and co-workers [13] have also identified a large proportion of Antarctic AD and KS OTUs not strongly matching to any known sequences in the MiBiG database, which is indicative of extensive novel chemistry being encoded in Antarctica soil microbiota. Interestingly, in Maxwell Bay samples, the percentage of unassigned sequences is higher for AD than the KS domain, while the opposite was observed by Borsetto [13], with a higher percentage of unassigned KS. This can be explained due to the different pair of primers used. While Borsetto and co-workers [13] used primers designed to target rare and divergent PKS and NRPS genes, in this study we used primers that were originally designed to amplify these domains from Actinobacteria [4,5].

#### 3.3.1. Metabolite/BGCs Tentatively Assigned to KS Domains

Among the metabolite/BGC matches of KS domain OTU sequences against the MiBIG database, the cytotoxic agents, BE-43547 A1-C2 [70] and rakidicins [71], which share a similar structure, were the two most abundant across all samples (Table 2 and Figure 8. Cyanobacterial BGCs encoding jamaicamide [72] and hapalosin [73] and the myxobacterial antimicrobial ajudazol [74] were also matched frequently. Each of these highly abundant matches seem to be associated to OTUs obtained from a specific bacterial phylum (Figure 8). Less abundant BGCs, such as those coding for the abyssomicins [75] and barbamide [76] seem to be matching OTUs distributed by a smaller number of samples, or to be sample-specific (Figure 8).

OTUs assigned to BGCs/metabolites with over 80% of identity were assessed in more detail, since they might indicate the presence of structural variants of such compounds.

Eighty-five different KS OTUs were assigned with over 80% of identity (Appendix A) to MiBIG metabolites. These are biosynthesized by four bacterial phyla (Actinobacteria, Cyanobacteria, Firmicutes, and Proteobacteria) and by the Ascomycota (Fungi) phylum (curiously, as mentioned above, fungi-associated KS domain OTUs were not found in BLAST searches). Over 40% of the matches belong to metabolites that have been isolated from *Streptomyces* species, also likely reflecting the fact that this hyper-prolific genus is strongly represented in the MiBiG database. The OTU assigned with the highest identity (90%) corresponded to spinosad, a non-antibiotic macrocyclic lactone with insecticide activity, isolated from a rare actinobacterial genus, *Saccharopolyspora* [77]. The second highest identity match corresponded to the fluvirucins, antifungal and antiviral molecules isolated from the actinobacterial genus *Actinomadura* [78].

Several other OTU sequences matched with over 85% identity to metabolites produced by *Streptomyces*, such as the recently discovered antibiotics niphimycins C–E [79] and the polyether ionophore antibiotic lasalocid [80], as well as the antifungal compound ibomycin [81].

#### 3.3.2. Assigned BGC/Metabolites to AD Domain

For the retrieved AD domain OTUs, the distribution of MiBIG-matched BGC/metabolites seems to be sample-specific as observed for amychelin and salinamide A (Figure 9) in sample TR1 and TC3, respectively. However, a few metabolites seem to be associated with all samples in high abundance (Table 3). This is the case of the three most abundant metabolites tentatively assigned to AD domains from our dataset (Table 3)—the phytotoxic cichopeptin [82], the lipopeptide glidopeptin [83], and the antibiotic lysobactin [84].

Nineteen different OTUs were assigned with over 80% of identity to the MiBIG database (Appendix A and Appendix A), corresponding to 13 different BGC/metabolites distributed by three bacterial phyla (Actinobacteria, Cyanobacteria, and Proteobacteria). The top match, with 96% identity to the BGC associated with the production of aurantimycin, was found in sample 66. Aurantimicyns, which are Gram-positive specific antibiotics but are also cytotoxic, were initially isolated from *Streptomyces aurantiacus* IMET 4391 [85]. The second and third hits were both associated to the BGC involved in the production of erythrochelin, a non-ribossomally assembled siderophore isolated from the actinobacterium, *Saccharopolyspora erythraea* [86]. Additional matches included pyoverdine [87], also a siderophore and viscosin, a biosurfactant [88] and antibiotic [89] NRP, which were both isolated from *Pseudomonas* strains.

Several OTU sequences matched BGCs/compounds produced by Cyanobacteria (Figure 9), including cyanopeptolins [90], puwainaphycins [91], nostopeptolides, and nostocyclopeptides [92]. Glidopeptin, matching with 80% of identity to highly abundant OTUs, was recently isolated and described to be encoded by a cryptic BGC from a Burkholderiales species [83].

## 4. Conclusions

Our study highlights the extensive unexplored chemical diversity at Maxwell Bay, Antarctica. We identified PKS and NRPS OTUs closely related to metabolites that have only very recently been described, are biosynthesized by cryptic genes or by uncultured and candidate phyla. Furthermore, a considerable fraction of the studied domains—most prominently AD domains—show very low identity to sequences within characterized BGCs. Hence, samples from this location show promise for the discovery of new natural products. In addition, despite being geographically very close to each other, the soil samples included in this study seem to contain a very distinct genetic repertoire, an observation that is relevant for planning bioprospection sampling campaigns. Despite the limited number of samples, collected in a short time period and in a limited geographic area, the approach used in this study has proved to be useful to detect both genes closely related to known biosynthetic compounds as well as those potentially linked to novel chemical diversity. A considerable proportion of matches to biosynthetic genes comes from rare and/or genera only recently described and that are usually recalcitrant to laboratory cultivation. Thus, new approaches to microbial cultivation [93] together with metagenomic strategies involving capture/heterologous expression of cryptic BGCs [94] will be useful to recover this hidden chemistry.

## Figures and Tables

**Figure 1 microorganisms-08-00279-f001:**
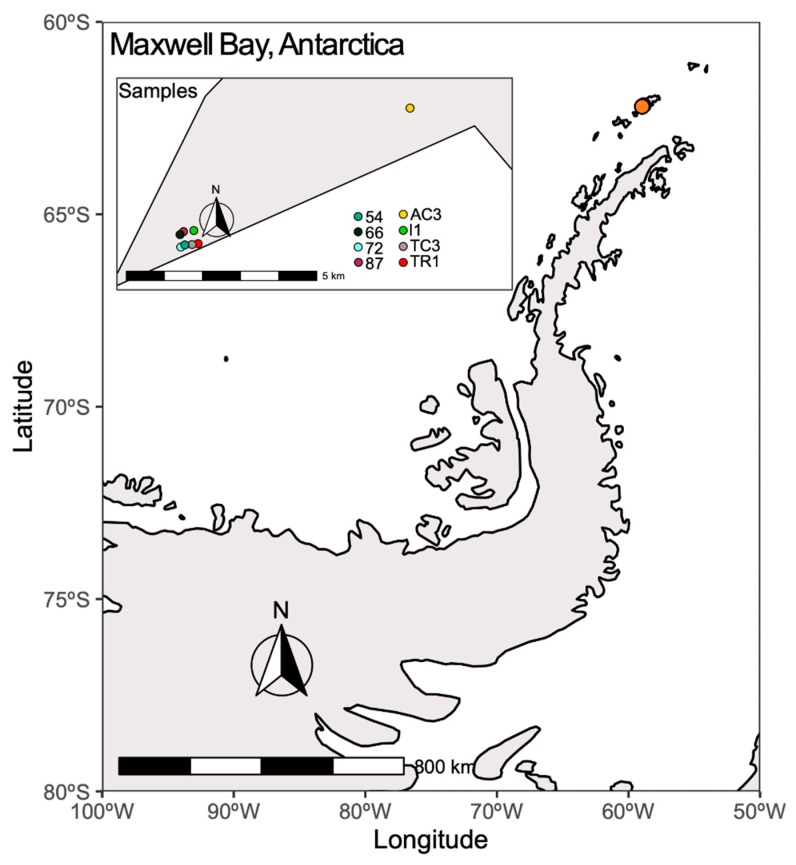
Location of sampling points in Maxwell Bay. This map was created using R v. 3.6.1 [26], R packages ggplot2 [27], sf [28], ggspatial [29], and rnaturalearth [30]. The map was edited using Inkscape v. 0.92.

**Figure 2 microorganisms-08-00279-f002:**
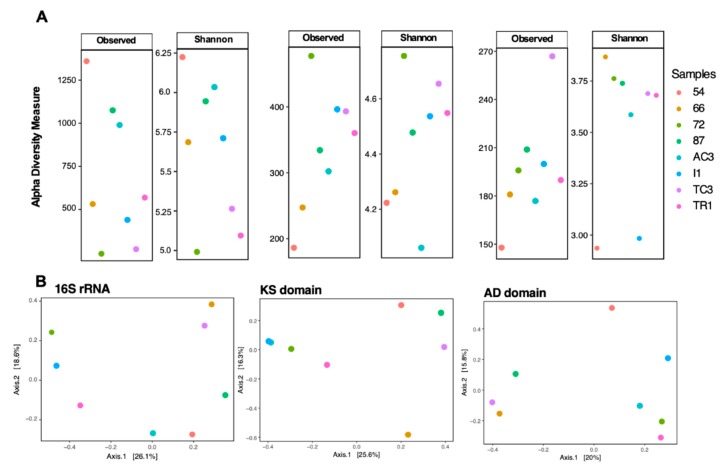
Alpha (**A**) and beta-diversity (**B**) metrics for 16S rRNA gene, ketosynthase (KS) and adenylation (AD) domains. Alpha diversity metrics used were number of amplicon sequence variants (ASVs) and operational domain units (OTUs) (grouped at 97% and in a second round at 95%) and Shannon diversity index.

**Figure 3 microorganisms-08-00279-f003:**
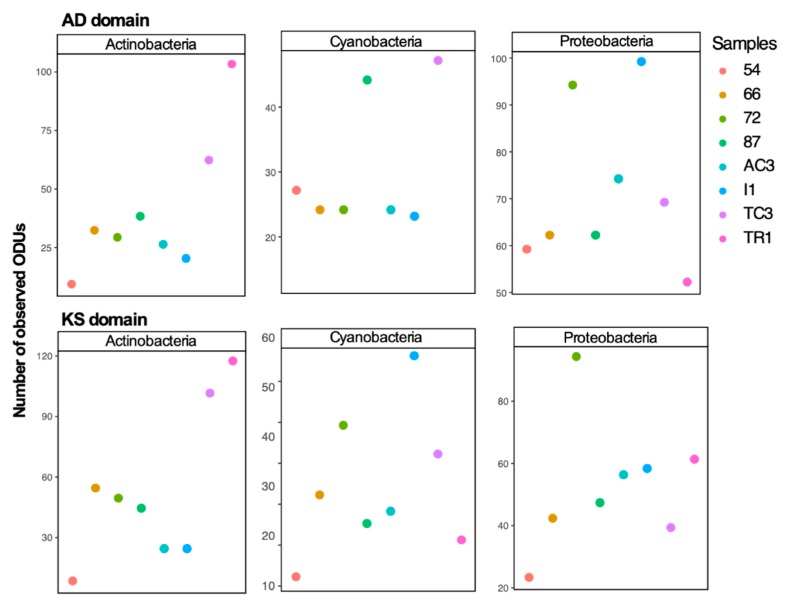
Alpha-diversity (number of OTUs grouped at 97% and in a second round at 95%) of KS and AD domains of Actinobacteria, Cyanobacteria, and Proteobacteria phyla.

**Figure 4 microorganisms-08-00279-f004:**
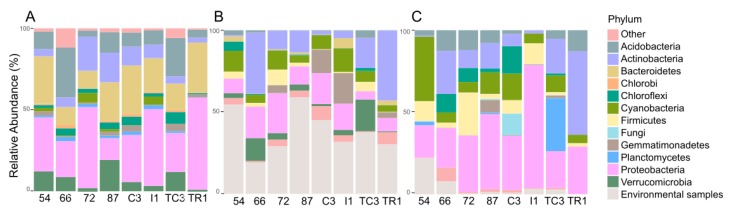
Relative abundance of the different phyla as revealed by 16S rRNA gene analysis (**A**) and the relative taxonomic provenience of KS (**B**) and AD (**C**) domains by identifying the best taxonomy match of the OTU biosynthetic domains using the NCBI database.

**Figure 5 microorganisms-08-00279-f005:**
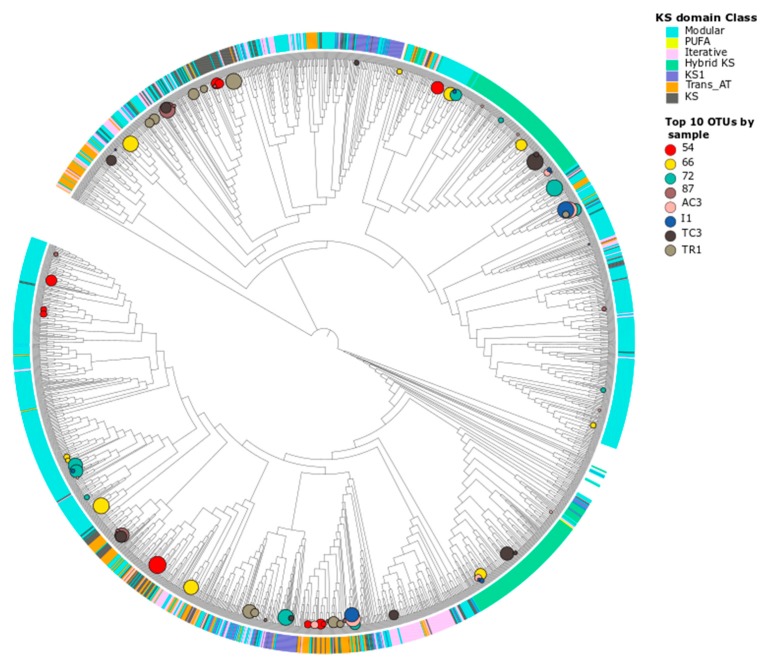
Maximum likelihood phylogenetic tree of KS domain OTU nucleotide sequences. The evolutionary history was inferred by using the maximum likelihood method based on the Tamura–Nei model and involved 1942 nucleotide sequences. KS classes assigned against NaPDOs database were included (color stripes) using iTOL as well as the relative abundances of the 10 most abundant OTUs by sample (represented in circles).

**Figure 6 microorganisms-08-00279-f006:**
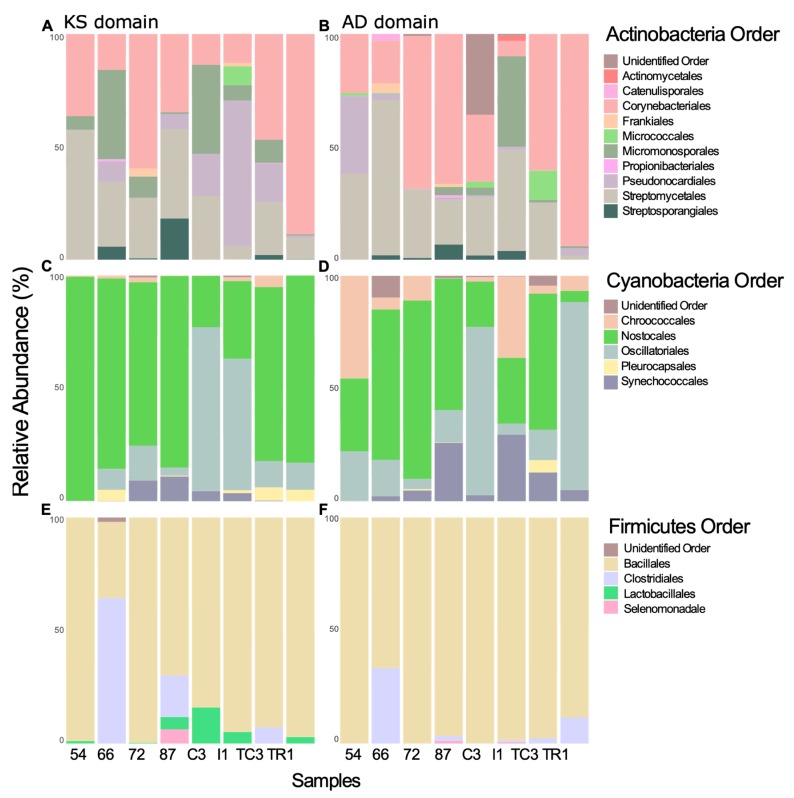
KS and AD domain taxonomic proveniences from Actinobacteria (**A**,**B**), Cyanobacteria (**C**,**D**), and Firmicutes bacterial phyla (**E**,**F**).

**Figure 7 microorganisms-08-00279-f007:**
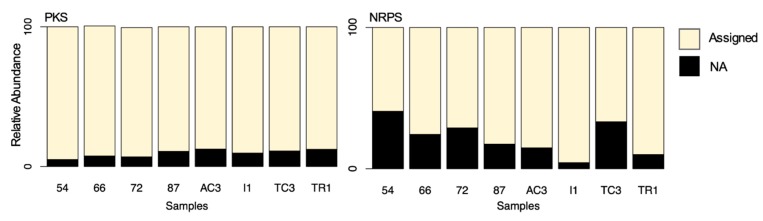
Distribution of assigned versus not-assigned polyketide synthase (PKS) and non-ribosomal peptide synthetase (NRPS) sequences by blast to the MiBiG database.

**Figure 8 microorganisms-08-00279-f008:**
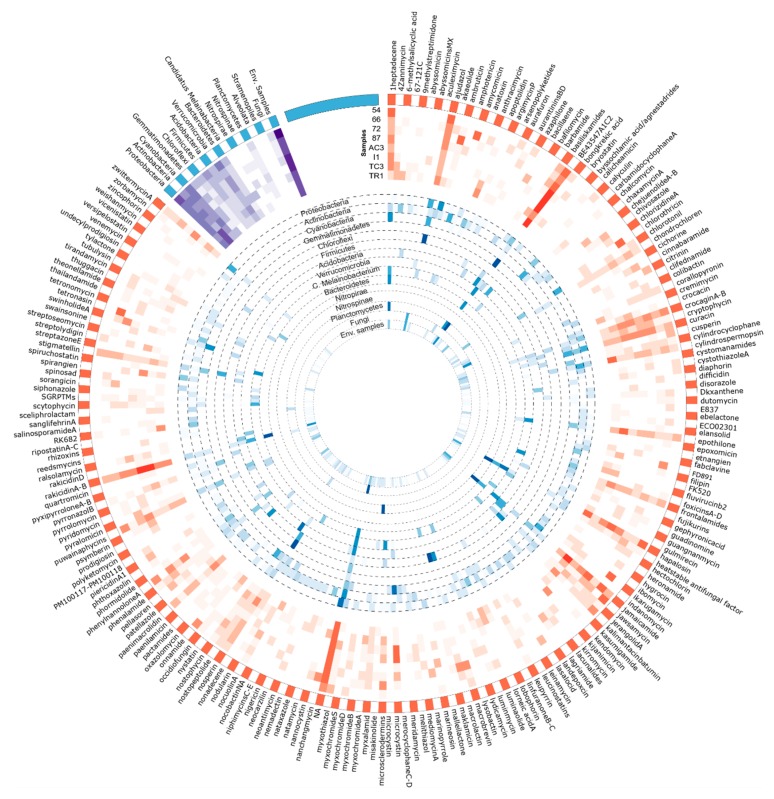
Circular visualization of the distribution of KS domain OTUs identified against the MiBIG database and its correspondent taxonomical provenience. Figure produced using circos software [45]. Read the figure clockwise, from the outside to the interior. The outside represents the metabolites assigned to the KS domain sequences by using the MiBIG database (orange and white). Heatmaps represent the sum of the metabolites assigned to the KS domain sequences for each sample (in red) and the relative abundance of its correspondent taxonomic provenience for each phylum (in blue). A third heatmap (in purple) represents the total number of metabolites from the KS domain with taxonomic provenience of each phyla per sample.

**Figure 9 microorganisms-08-00279-f009:**
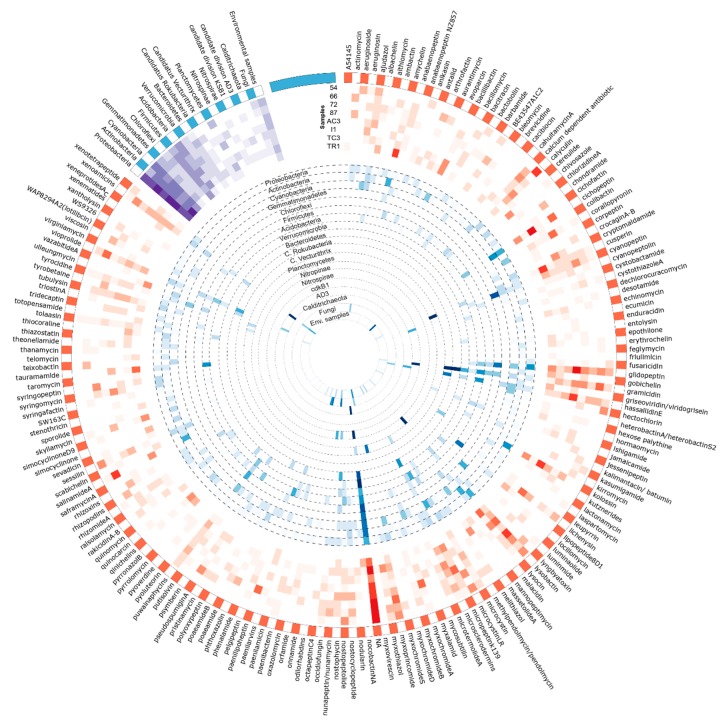
Circular visualization of the distribution of AD domain OTUs identified against MiBIG database and its correspondent taxonomical provenience. Figure produced using circos software [45]. Read the figure clockwise, from the outside to the interior. The outside represents the metabolites assigned to the KS domain sequences by using the MiBIG database (orange and white). Heatmaps represent the sum of the metabolites assigned to the AD domain sequences for each sample (in red) and the relative abundance of its correspondent taxonomic provenience for each phylum (in blue). A third heatmap (in purple) represents the total number of metabolites from KS domain with taxonomic provenience of each phyla per sample.

**Table 1 microorganisms-08-00279-t001:** Mantel correlation between the 16S rRNA gene diversity and the most abundant bacterial phyla 16S diversity with the AD and KS domain diversity.

	AD Domain		KS Domain	
	Mantel Statistic r	Significance	Mantel Statistic r	Significance
**16S rRNA**	0.6787	0.002*	0.7535	0.001*
**Phylum**				
**Proteobacteria**	0.6616	0.007*	0.3687	0.053
**Bacteroidetes**	0.5234	0.018*	0.3285	0.059
**Acidobacteria**	0.652	0.003*	0.4483	0.039*
**Actinobacteria**	0.5297	0.016*	0.5561	0.014*
**Verrucomicrobia**	0.6682	0.001*	0.5808	0.004*
**Chloroflexi**	0.6192	0.01	0.6378	0.005*
**Gemmatimonadetes**	0.4463	0.031*	0.3671	0.054
**Planctomycetes**	0.2158	0.121	0.4116	0.022*
**Cyanobacteria**	0.2158	0.102	0.1221	0.23

Statistically significant results were considered for *p* < 0.05 and are identified by * in the table.

**Table 2 microorganisms-08-00279-t002:** Distribution of the five most abundant MiBIG metabolites assigned to the KS domain sequences. The distribution (in relative abundance) of each metabolite by sample as well as the total distribution of the metabolite across all the samples is presented.

	Distribution in Each Sample (%)								
MiBIG Metabolite/BGC	54	66	72	87	AC3	I1	TC3	TR1	Total Distribution (%)
**BE-43547 A1-C2**	12.2	3.6	9.5	17.3	19.0	22.2	5.7	18.7	13.3
**NA**	5.0	6.9	7.5	10.7	12.5	9.6	11.1	12.3	9.8
**Rakicidin A/Rakicidin B**	2.3	1.9	4.0	3.4	14.1	16.5	2.7	2.5	6.0
**Jamaicamide**	2.1	8.6	0.7	12.0	3.3	1.4	13.3	0.8	5.0
**Hapalosin**	11.4	0.7	0.3	5.1	8.8	2.5	0.6	11.0	4.3
**Ajudazol**	10.6	5.7	3.5	3.9	1.8	1.1	2.4	5.4	3.7

**Table 3 microorganisms-08-00279-t003:** Distribution of the five most abundant MiBIG metabolites assigned to the AD domain sequences. The distribution (in relative abundance) of each metabolite by sample as well as the total distribution of the metabolite across all the samples is presented.

	Distribution in Each Sample (%)								
**MiBIG Metabolite**	54	66	72	87	AC3	I1	TC3	TR1	Total Distribution (%)
**NA**	40.7	24.3	28.9	17.4	14.8	4.3	33.3	10.0	20.5
**Cichopeptin**	0.0	0.0	9.1	0.0	0.1	29.8	0.0	2.7	7.5
**Glidopeptin**	2.0	4.3	1.4	3.6	30.4	2.3	0.8	4.4	6.0
**Lysobactin**	6.0	0.3	5.0	18.7	0.7	10.5	1.2	0.2	5.4
**Kolossin**	0.0	0.0	1.5	0.1	0.9	20.6	0.1	3.1	4.9
**Calyculin**	21.7	0.0	0.3	0.0	4.3	2.2	0.0	0.2	3.3

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
