# Peer review of "Diversity of Bacterial Biosynthetic Genes in Maritime Antarctica"

_microorganisms, 2020, doi:10.3390/microorganisms8020279_

Round 1

Reviewer 1 Report

This manuscript reports the sequencing analysis result for a total of 8 Antarctic samples collected from Maxwell Bay during February co on between 5 and 15 February 2014. The authors analyzed the bacterial 16S biomarker sequences and the KS and AD domain sequences of PKS and NPRS genes that are the biosynthetic basis of various bioactive natural products.  Generally, the manuscript presents a snapshot of the diversity of bacteria and the potential of natural products in Antarctic, which is relatively less studied compared to other continent areas.

The reviewer has the following concerns that needed to be addressed before publication.

The samples were collected from a relatively centering area and during a relatively centering period (February 5-15, 2014) although the samples are from 8 different locations. These locations are actually quite close. The data presented here only reflect the information in February.  The meaning of analyzing the difference in the biodiversity in 16S OUT and NP biosynthetic genes are actually limited. As mentioned above, these studied sites are very close. However, these sites might present the distinct features in their physiological and chemical features. Obviously, these physical-chemical or these routine soil parameters are important for understanding the spatial diversity difference, but missing. Although so, the significantly spatial differences in diversity of 16S and NP genes are clearly shown in the Figure 2 and Figure 4. For example, 72 and TR1 have the distinct 16S compositional patterns. The authors should discuss it. Additionally, in Figure 4b and 4c. the authors stated that they reflect the relative abundance pattern of KS (B) and AD (C) genes. This is quite confusing for the reviewer. According to the authors’ figure legends, the color codes for the different KS and AD genes should be shown in the column plot. However, in these two figures, the color codes reflect the different bacterial taxonomies (such cyanobacteria and Bacteroidetes ….). The is complete non-sense. This needs to be thoroughly revised. Fungi appeared in the figure 4. This study used the bacterial 16s biomarker to amplify the genes. How do you get the fungal sequences? Fungi is eukaryotic, NOT prokaryotic. The detailed comparison between the present and previous sequence analysis results are missing. What is the common finding shared in between this and previous work? What is the new finding of this manuscript? These should be highlighted in the abstract and conclusion.

Author Response

The authors truly acknowledge Reviewers for their effort and their suggestions that have contributed to improve the manuscript. Below, we addressed point-by-point each comment/suggestion made by each one of the two reviewers.

Reviewer 1

Point 1: The samples were collected from a relatively centering area and during a relatively centering period (February 5-15, 2014) although the samples are from 8 different locations. These locations are actually quite close. The data presented here only reflect the information in February.

Response 1: The authors agree with the reviewer that the study is both spatial and temporal limited in terms of sampling. According to this we have included in the conclusions section a sentence that states the sampling limitations of the present study.

Lines 433-436 (Track changes revised version):

Despite the limited number of samples, collected in a short time period and in a limited geographic area, the approach used in this study has proved to be useful to detect both genes closely related to known biosynthetic compounds as well as those potentially linked to novel chemical diversity.”

Point 2: The meaning of analysing the difference in the biodiversity in 16S OTU and NP biosynthetic genes are limited. 

Response 2: The authors understand the limitation of assessing a comparative analysis between the 16S OTUs and NP biosynthetic genes. According to this, we include information about the importance of assessing bacterial diversity (16S rRNA gene) while looking for biosynthetic genes diversity. Also,  a description of the methods used was included in the methods section on the new revised version.

Lines 127-137 (Track changes revised version):

“For the annotation of KS and AD sequences, the representative OTU sequences generated through USEARCH were aligned locally by BLAST+ version 2.9.0 [42] against the RefSeq non-redundant protein sequences (nr) database. The best hit of each representative KS and AD OUTU was retrieved, and BLAST results were curated in order to detect false positives, in which case these were removed. The accession number of each best hit was used to recover the lineage of the closest representatives of each KS and AD OTU using an in-house python script (https://github.com/antonioggsousa/get_taxonomy_lineage.py). The taxonomy classification was included in the respective KS and AD OTUs table generated in VSEARCH and imported to phyloseq to compute relative abundance of the taxonomic provenience of KS and AD domians that were represented in and bar plots.”

Lines 243-246 (Track changes revised version):

“The comparison between 16S rRNA and biosynthetic genes diversity taxonomy can provide clues on the ecological roles of the bacterial phyla-associated to the biosynthetic diversity and understand their distribution in the bacterial community.”

Point 3: As mentioned above, these studied sites are very close. However, these sites might present the distinct features in their physiological and chemical features. Obviously, these physical-chemical or these routine soil parameters are important for understanding the spatial diversity difference but missing. Although so, the significantly spatial differences in diversity of 16S and NP genes are clearly shown in the Figure 2 and Figure 4. For example, 72 and TR1 have the distinct 16S compositional patterns.  The authors should discuss it.

Response 3: We agree with the reviewer comment and in the revised version we include a principal component analysis (PCA) for the environmental variables (Figure S2 in supplementary material) alongside with a table with the variables measured (Table S1 in supplementary material). In addition information about the methods used were included in the methods section. In the “Results and Discussion” section results from this analysis were also presented. The differences between the 16S compositional patterns were also highlighted.

Lines 73-76 (Track changes revised version):

“Organic matter (OM) and water content were measured according with previous described methods [26]. Principal component analysis (PCA) was applied to the environmental variables measured in this study and also presented in a previous study [25], using R v. 3.6.1.”

Lines 189-191 (Track changes revised version):

“In agreement, the principal components analysis (PCA) applied to the environmental variables (Table S1) indicate that sample 54 was distinguished from others by being associated with high total organic matter and water content (Figure S2). 

Lines 208-216 (Track changes revised version):

“The bacterial taxonomy profile is quite similar across the studied samples, what is expected since samples were collected in a limited period spatial area. However, a few changes are noticed such as a relative increased of Proteobacteria and Bacteroidetes in sample TR1 in detriment of Actinobacteria and Verrucomicrobia (Figure 4A). According to the PCA analysis (Figure S2), TR1 diverge from the other samples by being related with higher concentrations of Hg, Pb, Zn and Cd in soils, thus these chemical variables may be driving differences in the taxonomical composition of this sample. The bacterial community distribution, including the most abundant bacterial phyla (Proteobacteria, Bacteroidetes, Acidobacteria and Actinobacteria) are in agreement with previous reports in Fildes Peninsula [21,24].

Point 4: Additionally, in Figure 4b and 4c. the authors stated that they reflect the relative abundance pattern of KS (B) and AD (C) genes. This is quite confusing for the reviewer . According to the authors’ figure legends, the color codes for the different KS and AD genes should be shown in the column plot. However, in these two figures, the color codes reflect the different bacterial taxonomies (such cyanobacteria and Bacteroidetes ….). The is complete non-sense.  This needs to be thoroughly revised. Fungi appeared in the figure 4.  This study used the bacterial 16s biomarker to amplify the genes. How do you get the fungal sequences? Fungi is eukaryotic, NOT prokaryotic.

Response 4: In the new version we changed the title of Figure 4 to clarify. Actually, figures 4B and 4C do not represent the relative abundance of KS and AD genes but the inferred taxonomic provenience of the recovered biosynthetic genes diversity.  Here the 16S data from our samples is only represented in figure A. In Figures B and C the  OTUs biosynthetic domain sequences were analysed by blast against NCBI database and the best taxonomy match was retrieved. The taxonomy of the best match was then associated to the biosynthetic domain to create a taxonomy table. This was done to determine the distribution of bacteria/fungi associated to the biosynthetic domains retrieved using specific KS and AD primer sets. We believe this is important to identify the most relevant intervenient phyla  and if the biosynthetic domains are associated to rare or candidate phyla. This information is important for example to direct culturing efforts in culturing studies. In the new version we also rephrase the methods related to this analysis to clearify.

Lines 131-137 (Track changes revised version):

“The accession number of each best hit was used to recover the lineage of the closest representatives of each KS and AD OTU using an in-house python script (https://github.com/antonioggsousa/get_taxonomy_lineage.py). The taxonomy classification was included in the respective KS and AD OTUs table generated in VSEARCH and imported to phyloseq to compute relative abundance of the taxonomic provenience of KS and AD domians that were represented in  bar plots.”

Lines 199-201 (Track changes revised version):

 “Figure 4 – Relative abundance of the different phyla as revealed by 16S rRNA gene analysis  (A) and the relative taxonomic provenience of KS (B) and AD (C) domains by identifying the best taxonomy match of the OTU biosynthetic domains using NCBI database.”

Lines 243-246: “The comparison between 16S rRNA and biosynthetic genes diversity taxonomy can give us a clue on the ecological role of the bacterial phyla-associated to the biosynthetic diversity and understand its distribution in the bacterial community. ”

Point 5: The detailed comparison between the present and previous sequence analysis results are missing.

Response 5: We found that the 16S rRNA gene taxonomic profile in our samples was very similar to previous described prokaryotic taxonomy profiles for the same region. In the new revised version we mention  that the our taxonomy were comparable with previous works.

 Lines 214-216 (Track changes revised version):

“The bacterial community distribution, including the most abundant bacterial phyla (Proteobacteria, Bacteroidetes, Acidobacteria and Actinobacteria) are in agreement with previous reports in Fildes Peninsula [21,24].”

Point 6: What is the common finding shared in between this and previous work? What is the new finding of this manuscript? These should be highlighted in the abstract and conclusion. 

Response 6:

With respect to the biosynthetic diversity we found that, similarly to other studies, that the biosynthetic diversity observed in Maxwell Bay (Table S2;S3) is clearly inferior to other environmental geographic areas (Lines 279-272). Although we show that rare bacterial phyla (such as Verrucomicrobia and Gemmatinomadetes) were concomitantly detected and play an important role in biosynthetic genes diversity. The main finding relies on the distribution of rare and candidate phyla found to account for a large proportion of biosynthetic genes diversity. We believe that this information is clear stated in the conclusion section of the manuscript.

Reviewer 2 Report

A research by Adriana Rego et al. Is devoted to the study of the genetic and biodiversity of Maxwell Bay, Antarctica. The research at a high experimental level was performed. The conclusions of the work presented by the authors are justified.

As a remark (wish), one can note the lack of the proposed chemical structures of novel chemical diversity, which were isolated from microorganisms of Maxwell Bay, Antarctica. These illustrations would be relevant here and would interest a wider group of researchers.

The work is of interest to researchers in the field of molecular biology and bioengineering. The Work can be taken with minor revision.

Author Response

The authors truly acknowledge Reviewers for their effort and their suggestions that have contributed to improve the manuscript. Below, we addressed point-by-point each comment/suggestion made by each one of the two reviewers.

Reviewer 2

Point 1: As a remark (wish), one can note the lack of the proposed chemical structures of novel chemical diversity, which were isolated from microorganisms of Maxwell Bay, Antarctica. These illustrations would be relevant here and would interest a wider group of researchers.

Response 1: The metabolites encoded by the MiBiG database BGCs that were matched with over 85% identity against our sequences were selected and its structures are now depicted - in total, 17  and 4 structures were drawn for KS and AD domain, respectively, and included in Figures S4 and S5 in supplementary material.  Methodological information with respect to this analysis was also included in the methods section (Lines 152).